# Optimizing Adsorption of 17α-Ethinylestradiol from Water by Magnetic MXene Using Response Surface Methodology and Adsorption Kinetics, Isotherm, and Thermodynamics Studies

**DOI:** 10.3390/molecules26113150

**Published:** 2021-05-25

**Authors:** Mengwei Xu, Chao Huang, Jing Lu, Zihan Wu, Xianxin Zhu, Hui Li, Langtao Xiao, Zhoufei Luo

**Affiliations:** 1College of Biological Science and Technology, Hunan Agricultural University, Changsha 410128, China; xumengwei0502@163.com (M.X.); chaoh@hunau.edu.cn (C.H.); hunauwuzihan@163.com (Z.W.); zhuxianxinstudent@126.com (X.Z.); lihui19981103@163.com (H.L.); 2Hunan Provincial Key Laboratory of Phytohormones and Growth Development, Hunan Agricultural Univesity, Changsha 410128, China; 3Technology Center of Changsha Customs, Hunan Key Laboratory of Food Safety Science & Technology, Changsha 410004, China; chinalulu139@126.com

**Keywords:** Fe_3_O_4_@Ti_3_C_2_ composite, 17α-ethinylestradiol, adsorption, response surface methodology, water remediation

## Abstract

Magnetic MXene composite Fe_3_O_4_@Ti_3_C_2_ was successfully prepared and employed as 17*α*-ethinylestradiol (EE2) adsorbent from water solution. The response surface methodology was employed to investigate the interactive effects of adsorption parameters (adsorption time, pH of the solution, initial concentration, and the adsorbent dose) and optimize these parameters for obtaining maximum adsorption efficiency of EE2. The significance of independent variables and their interactions were tested by the analysis of variance (ANOVA) and *t*-test statistics. Optimization of the process variables for maximum adsorption of EE2 by Fe_3_O_4_@Ti_3_C_2_ was performed using the quadratic model. The model predicted maximum adsorption of 97.08% under the optimum conditions of the independent variables (adsorption time 6.7 h, pH of the solution 6.4, initial EE2 concentration 0.98 mg L^−1^, and the adsorbent dose 88.9 mg L^−1^) was very close to the experimental value (95.34%). pH showed the highest level of significance with the percent contribution (63.86%) as compared to other factors. The interactive influences of pH and initial concentration on EE2 adsorption efficiency were significant (*p* < 0.05). The goodness of fit of the model was checked by the coefficient of determination (R^2^) between the experimental and predicted values of the response variable. The response surface methodology successfully reflects the impact of various factors and optimized the process variables for EE2 adsorption. The kinetic adsorption data for EE2 fitted well with a pseudo-second-order model, while the equilibrium data followed Langmuir isotherms. Thermodynamic analysis indicated that the adsorption was a spontaneous and endothermic process. Therefore, Fe_3_O_4_@Ti_3_C_2_ composite present the outstanding capacity to be employed in the remediation of EE2 contaminated wastewaters.

## 1. Introduction

The synthetic estrogenic steroid 17α-ethynilestradiol (EE2), the active ingredient of most contraceptive medicine, has been widely used to adjust the animal or human pregnancy and reproduction [1]. The treated effluent from animal excrete always contains EE2 and makes the effluents become a major pathway for introducing EE2 into the aquatic environment [2]. EE2 has been widely distributed in surface waters with a detectable concentration in the world [3,4]. Numerous studies have reported the ability of EE2 to alter sex determination, delay sexual maturity, and decrease the secondary sexual characteristics of exposed organisms [5,6]. With its ubiquitous occurrence and high endocrine disrupting potency, EE2 has become a widespread problem in the aquatic environment [7]. Therefore, it is highly desirable to remove the EE2, in particular, from water or wastewater.

Owing to the high resistance of EE2 to degradation, many research efforts have turn to physical treatments by using adsorbent materials for the removal [8]. MXene Ti_3_C_2_ was two-dimensional and has a micro-crack structure, which has a large specific surface area, high porosity, and good stability [9]. The MXene Ti_3_C_2_ nanosheet is composed of transition metal nitrides, carbides, or carbonitrides. Its general formula is Mn + 1XnTx (*n* = 1–3), where “M” represents the transition metal element, “X” is C or N or CN, and Tx stands for the terminal group –OH or –F, eta [10]. Ti_3_C_2_ has been demonstrated to adsorb a variety of environmental pollutants, including organic dyes and heavy metal ions [11]. Magnetic Ti_3_C_2_ could exhibit high adsorption efficiency and be conveniently recycled by an external magnetic field. To our knowledge, there is no literature about the use of magnetic Ti_3_C_2_ to the enrichment of EE2. The application of magnetic Ti_3_C_2_ material as adsorbents to water remediation need to be further explored.

It is important to investigate the interactive effects between the process variables and optimize the adsorption parameters in the liquid–solid interface adsorption process. The single-factor experiment is a conventional approach for the optimization of adsorption variables, it requires a large number of experiments; consequently, it is laborious and time-consuming [12]. Response surface methodology (RSM) is a collection of statistical and mathematical techniques, which could effectively evaluate the responses influenced by multiple parameters and optimize the complex process [13]. Compared with the conventional statistical strategy, RSM could reduce the cost, decrease the number of experiments, and need less time [14]. RSM is based on multivariate statistics including experiment design, process optimization, and statistical model [15], and it has widely been applied in adsorption process optimization [16]. Box–Behnken designs (BBD) is classified as response surface design that consists of a central point and the middle points of the edges of the circle circumscribed on the sphere [17]. BBD is recommended owing to its advantage in the application of magnetic Ti_3_C_2_ to adsorb EE2.

In this study, the magnetic nanocomposite Fe_3_O_4_@Ti_3_C_2_ was prepared, characterized, and used as an absorbent for the removal of EE2 from aqueous solution. The interactive effects of the operating parameters (adsorption time, pH of the solution, initial concentration, and the adsorbent dose) were investigated. The parameters affecting the adsorption efficiency were optimized by RSM. The second-order polynomial equation provided an excellent explanation of the relationship between the response (EE2 adsorption efficiency) and these independent parameters. Moreover, the study of adsorption isotherms, kinetics, and thermodynamics were also carried out to explore the adsorption mechanism.

## 2. Results

### 2.1. Characterization of Fe_3_O_4_@Ti_3_C_2_ Composite

Figure 1a illustrates the X-ray diffraction (XRD) patterns of the Fe_3_O_4_@Ti_3_C_2_ nanocomposites. The characteristic diffraction peak (002) of the material fit well with Ti_3_C_2_, which was consistent with the report [18]. The characteristic diffraction peaks (220, 311, 511, 440) of Fe_3_O_4_ match well with the standard XRD data of magnetite [19]. A magnetic hysteresis curve shows that the saturation magnetization of Fe_3_O_4_@Ti_3_C_2_ was 34.8 emu g^−1^ in Figure 1b. TGA was used to identify the thermal stability of the prepared Fe_3_O_4_@Ti_3_C_2_. Figure 1c show a comparison of the weight losses between the prepared Fe_3_O_4_@Ti_3_C_2_ upon heating under nitrogen and air atmosphere from room temperature to 800 °C. At low temperature, it may be caused by the evaporation of free water adsorbed on the surface of Fe_3_O_4_@Ti_3_C_2_ nanomaterials. At higher temperature, it may be caused by the functional groups of Fe_3_O_4_@Ti_3_C_2_. The less weight loss indicates that Fe_3_O_4_@Ti_3_C_2_ show remarked thermogravimetric stablility. N_2_ adsorption–desorption isotherms of composite material (Figure 1d) referred to the type IV hysteresis hoop (IUPAC). The specific surface area of composite material was calculated to be 19.38 m^2^ g^−1^ by BET analysis. Figure 1e,g illustrates the typical Ti_3_C_2_ MXene with layered structure. Figure 1f,h displays the distribution of Fe_3_O_4_ nanoparticles on the surface of multi-layered Ti_3_C_2_ MXene.

A mathematical model was formed to describe the effect of process variables and predict the response of the independent variables. The response variable Y (adsorption % of EE2 by magnetic adsorbent from aqueous solution) can be expressed as Y = f (A, B, C, D), where A, B, C, and D are the coded values of the four process variables. The selected relationship being a second-degree response surface was expressed as below in Equation (1):(1)$$Y=\beta_{0}+\sum_{i=1}^{n}\beta_{i}X_{i}+\sum_{i=1}^{n}\beta_{ii}X_{ii}^{2}+\sum \beta_{ij}X_{i}X_{j}+e$$
where Y is the dependent variable, β_0_ is the model constant, β_i_ is the linear coefficients, β_ii_ is the quadratic coefficients, β_ij_ is the interaction coefficients, and X_i_, X_j_ are the coded values of the independent process variables, and e is the error [20]. Analysis of the experimental design data and calculation of predicted responses were carried out by Design Expert 8.0.6 software.

### 2.2. Optimum of Adsorption Condition by Response Surface Methodology

The response values at different experimental combinations for coded variables are listed in Table 1. The adsorption efficiency of EE2 from the water using magnetic Ti_3_C_2_ nanocomposite ranged from 63.19% to 92.23%.

By applying multiple regression analysis on the experimental data, the response variables and the test variables were related by the following second-order polynomial Equation (2):Y (%) = 90.45 + 2.28A + 5.12C + 0.98D − 1.44AB + 1.03AC − 1.4AD + 1.05BC + 0.16BD − 1.75CD − 5.01A^2^ − 17.15B^2^ − 3.39C^2^ − 2.28D^2^(2)

The quadratic model was used to evaluate the influence of the process variables on the EE2 adsorption (in percent) from the water using the magnetic Fe_3_O_4_@Ti_3_C_2_.

The significance of the independent variables and their interactions were tested by the analysis of variance (ANOVA). The *p* value is used to check the significance of the coefficient. The ANOVA results (Table 1) suggest that the model was highly significant with a high F value (F_model_ = 62.16) and a very low probability *p*-value (*p* < 0.0001) [21]. The nonsignificant *p*-value (0.09) for the lack of fit indicates that the model could adequately fit the experiment data [22]. As shown in Table 1, the *p*-value of B, C, D, BC, A^2^, B^2^, C^2^, and D^2^ are all less than 0.05, which indicates that these variables are significant and have an influence on EE2 adsorption. It is evident that all the linear terms except A (time) are statistically the most significant factors (*p* < 0.0001). Moreover, the statistical results suggest that only the interaction of pH and concentration is statistically significant (*p* < 0.05). Among the quadratic terms, the pH term is the most significant (*p* < 0.001), so the solution pH exhibits a highly significant effect on the adsorption of EE2 from the water.

Figure 2a shows the predicted versus actual values of the adsorption capacities, indicating that the actual values were distributed relatively close to the straight line. The quadratic model was the requisite for predicting the efficient adsorption of EE2 under the experimental parameters studied. The high coefficient of determination R^2^ (0.9719) of the equation is close to 1, meaning a high degree of correlation between the observed and predicted values. Moreover, a closely high value of the adjusted coefficient of determination $R_{adj}^{2}$ (0.9437) also showed a high significance of the model. The plot of studied residuals versus run number was tested and displayed in Figure 2b. The residuals were scattered randomly from +3 to −3, indicating that the experimental data were well fitted with the model.

### 2.3. Effects of Model Factors and Their Interactions on EE2 Adsorption

The significance of the quadratic model coefficients was evaluated by the Student’s t-test, and the results are listed in Table 2. The t-value is the ratio of the estimated parameter effect and the estimated parameter standard deviation. For the regression coefficients, a positive sign of the coefficient represents a synergistic effect, while a negative sign indicates an antagonistic effect. The interactive terms of AD (time versus dose), BC (pH versus concentration), BD (pH versus dose), and CD (concentration versus dose) showed a positive significant effect on the process, whereas the interactive terms of AB (time versus pH) and AC (time versus concentration) exhibited a negative relationship with the EE2 adsorption process. The percent contribution (PC) of each individual term in the model was calculated using the sum of squares (SS) values of the corresponding term. The PC of a term is obtained as the ratio of SS of an individual term to that of the sum of SS for all the terms, as followed in Equation (3):(3)$$\mathrm{PC}=\frac{\mathrm{SS}}{\sum SS}\times 100.$$

As shown in Table 2, the pH (A) showed the highest level of significance with PC (63.86%) as compared to other factors, and it was followed by the quadratic terms (23.06%). The pH of the solution played an important role in the EE2 adsorption process.

### 2.4. Three-Dimensional (3D) Response Surface and Contour Plots

The three-dimensional (3D) response surface and contour plots were constructed based on the quadratic model. The influence of four different variables on EE2 adsorption was visualized in the 3D response surface and contour plots. The elliptical contour plot in Figure 3a suggests that the interactive effects of time and pH on EE2 adsorption were significant [23]. Similar counter plots were also observed in Figure 3b,d–f. There were significant interactive effects on EE2 adsorption of time and concentration, pH and concentration, pH and dose, as well as concentration and dose [24]. The contour lines in Figure 3c presented a continuous rounded shape, implying that the interaction of time and dose was not significant [25].

### 2.5. Adsorption Kinetics

To study the effect of contact time on the efficiency of the adsorption process, batch adsorption experiments were conducted at a fixed adsorbent dose 88.9 mg L^−1^, pH 6.4, and initial EE2 concentration of 0.98 mg L^−1^ while varying the contact time from 0.5 to 10 h. The pseudo first-order and pseudo second-order models were separately applied to fit these experimental data in an attempt to explain the adsorption kinetic of EE2 onto Fe_3_O_4_@Ti_3_C_2_. Figure 4 showed the effect of contact time on the adsorption of EE2 onto Fe_3_O_4_@Ti_3_C_2_. The adsorbed amount of EE2 increased dramatically in the first 0.5 h and then increased slowly from 0.5 to 4 h, and even an increase to 10 h was slight.

The pseudo first-order model is the empirical kinetic equation for one-site occupancy adsorption, which simulates a rapid adsorption due to the absence of sorbate–adsorbate interaction [26]. The pseudo second-order model involves the potential adsorption procedures, such as surface adsorption, external film diffusion, and intra-particle diffusion [27]. The equation of the pseudo first-order model and the pseudo second-order model isotherm are described in Table 3.

### 2.6. Adsorption Isotherms

To study the effect of initial EE2 concentration on the efficiency of the adsorption process, adsorption isotherm tests were conducted for EE2. Batch adsorption experiments were conducted at a fixed 88.9 mg L^−1^ adsorbent dose, pH 6.4, and the contact time of 24 h, while varying the initial concentration of EE2 from 0.05 to 2 mg L^−1^.

Measured and modeled adsorption isotherm data for EE2 are shown in Figure 5. According to Figure 5, it is clearly observed that the adsorption capacity increases from 0.05 to 3.83 mg g^−1^ with the increasing EE2 initial concentration. A higher EE2 concentration leads to a higher driving force for mass transfer from the bulk solution to the adsorbent surface, resulting in a higher adsorption capacity. The corresponding isotherm parameters obtained by nonlinear regression analysis are summarized in Table 4.

### 2.7. Adsorption Thermodynamics

Thermodynamic study is of utmost importance for the proper prediction of the adsorption mechanism. The effect of temperature on the EE2 removal by Fe_3_O_4_@Ti_3_C_2_ was explored at different temperatures (288, 298, 308, 318 K) under optimized conditions. The thermodynamic parameters of EE2 adsorption onto Fe_3_O_4_@Ti_3_C_2_ were graphically determined according to the thermodynamic laws. △G° of the adsorption processes can be determined by the classical Van’t Hoff equation below:(4)$$\Delta G{}^{\circ}=-\;RTlnk_{d}$$
(5)ΔG°=ΔH°−TΔS°
(6)$$lnk_{d}=\frac{\Delta S{}^{\circ}}{R}-\frac{\Delta H{}^{\circ}}{RT}$$
(7)$$\Delta H{}^{\circ}=R\frac{dlnK}{d(1/T)}$$
where R is the universal gas contant (8.314 J mol^−1^ K^−1^), T is the absolute temperature (K), K_d_ is the adsorption equilibrium constant, ΔS is the change in entropy (kJ mol^−1^ K^−1^), and ΔH is the change in heat of adsorption (kJ mol^−1^) at a constant temperature. The plot of lnK versus 1/T is presented in Figure 6. The values of various thermodynamic parameters are listed in Table 5. It is observed from Table 5 that ΔH and ΔS are positive, and ΔG is negative in adsorption processes.

## 3. Discussion

According to the Design Expert 8.0.6 software and the regression model, the predicted optimal condition for EE2 adsorption by Fe_3_O_4_@Ti_3_C_2_ was suggested as following: adsorption time 6.7 h, pH of the solution 6.4, initial EE2 concentration 0.98 mg L^−1^, and the adsorbent dose 88.9 mg L^−1^, to achieve the maximum adsorption of EE2 (97.08%). The corresponding experimental value of the EE2 adsorption efficiency under the optimum condition was 95.34%, which is very close to the predicted optimum value. Compared with the single-factor experiment, it is clearly showed that the optimal conditions predicted by RSM were accurate and reliable.

Figure 3a shows the three-dimensional response surfaces representing the effects of interaction terms of time and pH with constant initial concentration and adsorbent dose. The EE2 adsorption efficiency (percentage) increased as the time increased until it achieved equilibrium. The effect of the initial solution pH on EE2 adsorption was investigated in the range from 4 to 10. In view of the presence of hydroxyl groups in adsorbent and the skeleton of Fe_3_O_4_@Ti_3_C_2_, it could be speculated that hydrophobic forces and hydrogen bonds would be the predominant driving force during adsorption [28]. Thus, the EE2 adsorption is favored under moderate pH conditions [29].

The interactive effect of time and initial EE2 concentration at constant pH and adsorbent dose is shown in Figure 3b. The adsorption efficiency increases with an increase of both time and concentration of EE2 within their respective experimental ranges. The concentration dependence may be owing to the number of binding sites on the adsorbent surface [30]. Figure 3c showed the interaction of time and adsorbent dose and their relation on adsorption efficiency. The adsorption efficiency was rapidly increased in the early stages and gradually slowed down until equilibrium. It was noted that the number of active adsorption sites on the surface increased at high adsorbent dose [31]. Figure 3d illustrated the interactive influence of pH and concentration on EE2 adsorption. It was evidenced that the EE2 adsorption increased with increasing EE2 concentration, while the change of adsorption efficiency was not significant over a broad range of concentrations in the adsorption system. Figure 3e showed the combined effect of the solution pH and the adsorbent dose. It was suggested that the EE2 adsorption increased with the increase of adsorbent dose. The combined effect of initial concentration and adsorbent dose is shown in Figure 3f. It could be seen that the EE2 adsorption increased more remarkably as the EE2 concentration increased. It may be attributed to the fact that increasing the EE2 concentration would increase the relatively more active binding sites [32].

The pseudo second-order model fit well with the measured adsorption data with the high R^2^. Q_e_ values calculated with the pseudo second-order model were close to the measured data, which indicated that the sorbet interaction may be dependent on the amount of the solute adsorbed on the surface of the adsorbent. The Langmuir and Freundlich isotherm models help to understand the mechanism of adsorption. The Langmuir adsorption model is the theoretical formula of adsorption based on a strong specific interaction between sorbent and adsorbent that only occurs on monolayer coverage [33]. The adsorption data for EE2 fit well with Langmuir with high correlation coefficient R^2^, indicating that the adsorption mainly is monolayer coverage. The positive △H value confirms the endothermic nature of the overall sorption process, while the positive value of △S proved a good affinity between the EE2 molecules and the Fe_3_O_4_@Ti_3_C_2_ adsorbent surface, and a high randomness at the solid/solution interface with some structural changes during the adsorption process. Additionally, the negative △G indicates the feasibility and spontaneity of the adsorption process [19].

Adsorption is a mixed process influenced by properties of adsorbent and adsorbate. The EE2 has the acid dissociation constant (pKa ≈10.33) [34]. Eletrostatic interaction could be negligible on EE2, which is non-inoizable under these experiment pH conditions. Meanwhile, MXene possesses higher hydrophilicity without aromatic rings, and hydrophobic partitioning cannot be considered in this study. In view of the presence of aromatic rings and phenolic hydroxyl groups in EE2, and the terminals of MXene Ti_3_C_2_ that consist of -OH, -O, and/or the -F surface, it could be speculated that hydrogen bonding would be the main driving force during adsorption. The adsorption capacity of Fe_3_O_4_@Ti_3_C_2_ was compared with other adsorbents reported previously, as shown in Table 6, and it can be seen that the sorption capacity of Fe_3_O_4_@Ti_3_C_2_ is higher than those of other adsorbents. Moreover, Fe_3_O_4_@Ti_3_C_2_ shows good magnetic properties and thus shows promise as a novel adsorbent for EE2 removal from water.

## 4. Materials and Methods

### 4.1. Materials

EE2 (≥98%) was purchased from Sigma-Aldrich Co. (St. Louis, MO, USA), and its stock solution (1000 mg L^−1^) was prepared with methanol. *N*, *O*-Bis (trimethylsilyl) trifluoroacetamide (BSTFA) with 1% trimethylsilyl chloride (TMCS, >99%) was purchased from Regis Technologies (Morton Grove, IL, USA). Methanol (≥99.9%, Sigma-Aldrich, St. Louis, USA) and other solvents were all of at least analytical grade. An SPE C18 column (Waters Corp., Milford, MA, USA) was used for chromatographic separation. Ultra-pure water (resistivity ≥18.25 MΩ cm^−1^) was obtained from a WaterPro water system (Beijing, China). The pH of the solution was adjusted with 0.1 M NaOH or 0.1 M HCl solution and monitored using a pH meter (pHSJ-3F, JK, China).

### 4.2. Preparation and Characterization of Fe_3_O_4_@Ti_3_C_2_

Ti_3_AlC_2_ powder (>98 wt % purity) was obtained from 11 Technology Co., Ltd. (Jilin, China). Fe_3_O_4_@Ti_3_C_2_ was synthesized according to the hydrothermal method. A detailed characterizations of the X-ray diffraction (XRD), vibrating sample magnetometer (VSM), scanning electron microscopy (SEM), and transmission electron microscopy (TEM) were provided.

### 4.3. Adsorption Experiments and Analytical Method

Considering the practicality of wastewater treatment, the adsorption experiments were performed at room temperature. In each experiment, a certain amount of Fe_3_O_4_@Ti_3_C_2_ was added to a glass conical flask loaded with 50 mL aqueous EE2 solution of a certain concentration. The flask was kept in a water bath shaker at room temperature for a set period of time. The adsorbed EE2 was desorbed with 5 mL methanol after shaking for 30 min. The concentration of EE2 was determined by gas chromatography-mass spectrometry (GC-MS), and the sample pretreatment was using the solid-phase extraction (SPE) method. More detailed descriptions of the analytical procedure are given in the previous study [19]. The EE2 adsorption (in percent) was calculated below in Equation (8):(8)$$\mathrm{Adsorption}(\%)=\frac{C_{0}{-\;C}_{f}}{C_{0}}\text{×}100$$
where C_0_ and C_f_ are the initial and final concentration of EE2 in the solution, respectively. All sorption experiments were performed in duplicate. The super paramagnetic properties of prepared material can satisfy their fast separation from aqueous dispersion within 30 s by using an external magnetic field.

### 4.4. Box–Behnken Design

Four factors, i.e., adsorption time, pH of the solution, initial concentration of EE2, and adsorbent dose were chosen. According to the principle of BBD, after definition of the range of each of the process factors, these factors were prescribed into three levels, coded −1, 0, and +1 for low, intermediate, and high value, as shown in Table 7, respectively. A total of 29 experiments were performed in a randomized order.

## 5. Conclusions

In this study, the magnetic composite Fe_3_O_4_@Ti_3_C_2_ was prepared and used as adsorbent to remove EE2 for water environment remediation. The proposed BBD approach provided a critical analysis of the interactive influences of the selected variables on the EE2 adsorption process of adsorbent. pH was the most significant parameter in the EE2 adsorption process. The interactive influence of pH and initial concentration was significant. The model predicted values were in good agreement with the experimentally determined values. The optimum process conditions for the maximum adsorption of the EE2 were identified. The maximum adsorption efficiency of the experiment value was found to agree closely with the model predicted value. The RSM approach successfully reflects the impact of various factors, and the established model well agrees with the actual situation. Kinetics data suggested that the EE2 adsorption process on Fe_3_O_4_@Ti_3_C_2_ was predominant by the pseudo-second-order adsorption mechanism. The adsorption experimental data were well described by the Langmuir model. Thermodynamic study showed that the adsorption process was spontaneous and endothermic. The prepared Fe_3_O_4_@Ti_3_C_2_ showed good absorption efficiency of EE2 water solution, indicating its promise for practical applications in environmental remediation.

## Figures and Tables

**Figure 1 molecules-26-03150-f001:**
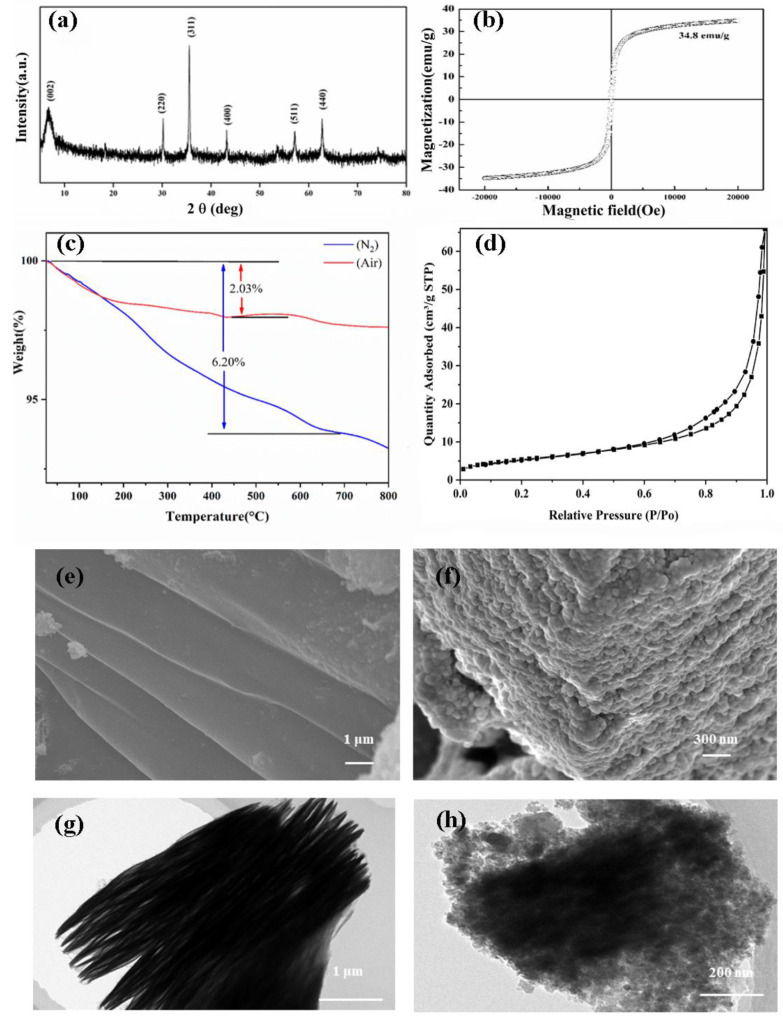
XRD spectrum (**a**); magnetization hysteresis loop (**b**); TGA thermograms of Fe_3_O_4_@Ti_3_C_2_ (**c**); BET analysis of Fe_3_O_4_@Ti_3_C_2_ (**d**), SEM image of Ti_3_C_2_ (**e**); Fe_3_O_4_@Ti_3_C_2_ (**f**); TEM image of Ti_3_C_2_ (**g**), and Fe_3_O_4_@Ti_3_C_2_ (**h**).

**Figure 2 molecules-26-03150-f002:**
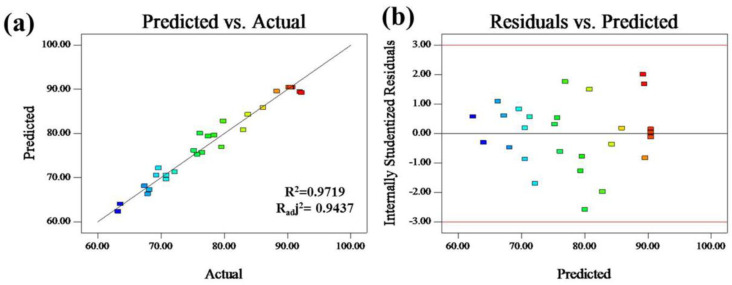
Plot of the measured and predicted values of the response variable (**a**); plot of Studentized residuals versus experimental run number (**b**).

**Figure 3 molecules-26-03150-f003:**
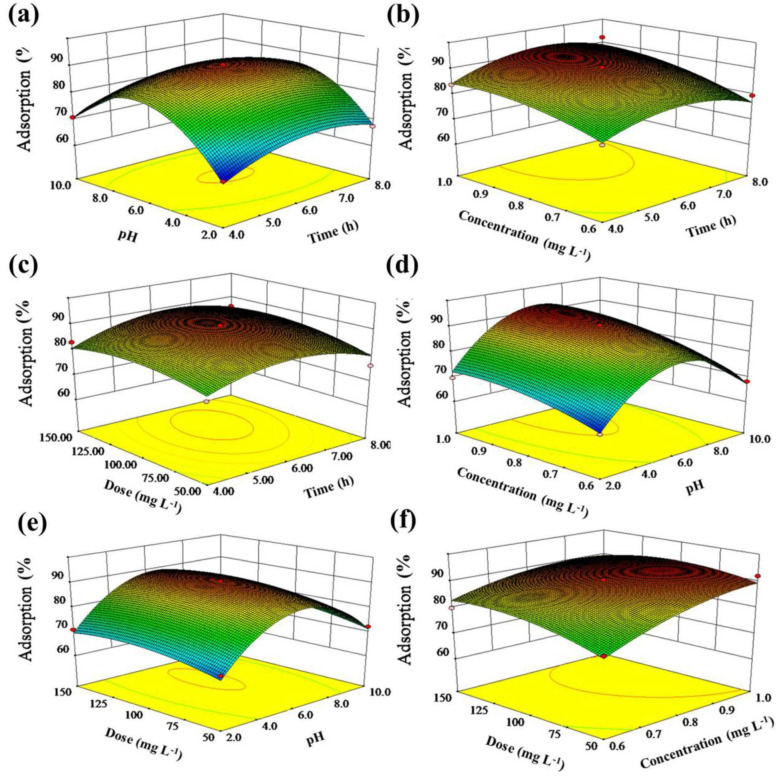
Three-dimensional response surface plot for the effects of time and pH (**a**); time and concentration (**b**); time and dose (**c**); pH and concentration (**d**); pH and dose (**e**); concentration and dose (**f**).

**Figure 4 molecules-26-03150-f004:**
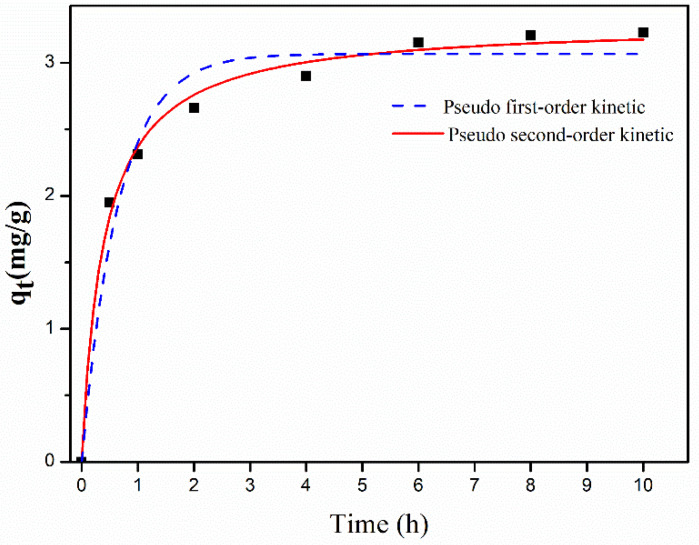
Kinetic models for the adsorptions of EE2 onto Fe_3_O_4_@Ti_3_C_2_ (C_0_ = 0.98 mg L^−1^, dose 88.9 mg L^−1^, pH 6.4, time 0.5 to 10 h).

**Figure 5 molecules-26-03150-f005:**
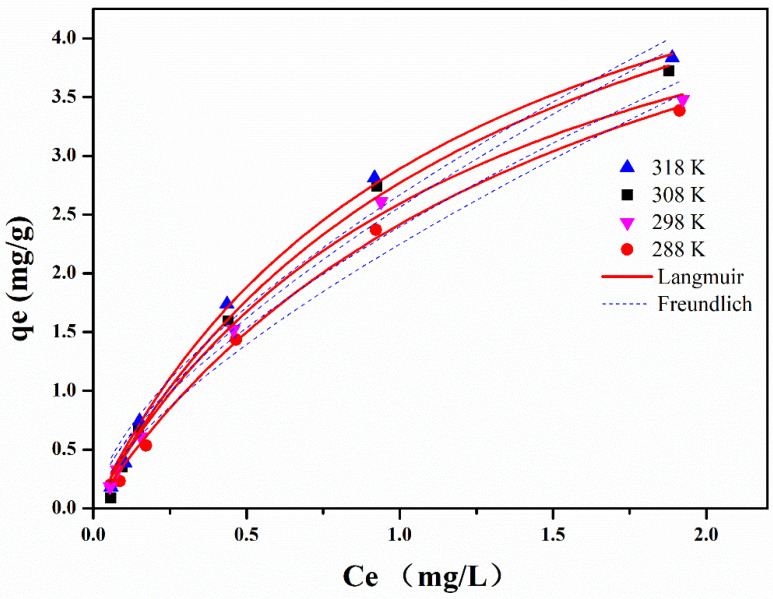
Adsorption isotherm data for EE2 on Fe_3_O_4_@Ti_3_C_2_ (C_0_ = 0.98 mg L^−1^, dose 88.9 mg L^−1^, time 24 h, pH 6.4, and temperature 288–318 K).

**Figure 6 molecules-26-03150-f006:**
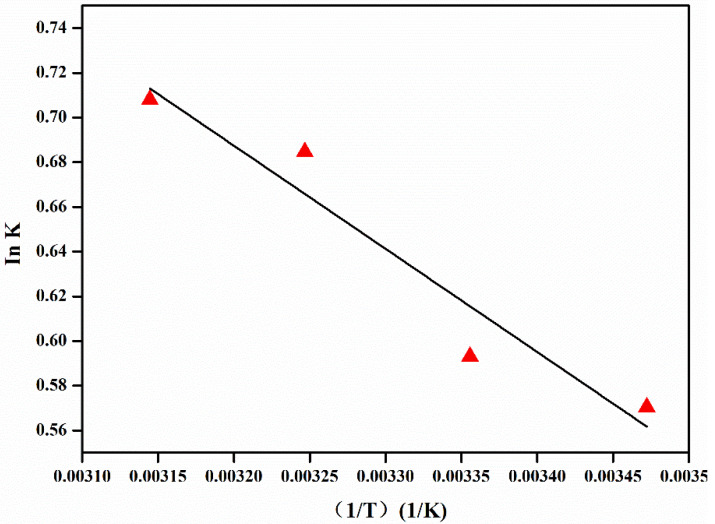
Van’t Hoff plot for the determination of ΔH°, ΔS°, and ΔG° (C_0_ = 500 μg L^−1^).

**Table 1 molecules-26-03150-t001:** Analysis of variance (ANOVA) for the regression equation.

Source	Sum of Squares	Degrees of Freedom	Mean Sum of Squares	*F* Value	*p* Value
Model	1099.86	14	78.56	62.16	<0.0001
A-Time	7.21	1	7.21	3.11	0.0003
B-pH	65.43	1	65.43	30.70	<0.0001
C-Concentration	129.69	1	129.69	56.00	<0.0001
D-Dose	79.00	1	79.00	34.11	<0.0001
AB	8.29	1	8.29	3.58	0.0793
AC	11.83	1	11.83	5.11	0.0603
AD	7.34	1	7.34	3.17	0.0967
BC	4.41	1	4.41	1.90	0.0493
BD	0.67	1	0.67	0.29	0.5985
CD	1.01	1	2.500 × 10^−5^	1.079 × 10^−5^	0.9974
A^2^	16.19	1	16.19	6.99	0.0193
B^2^	489.30	1	489.30	211.27	<0.0001
C^2^	56.20	1	56.20	24.27	0.0002
D^2^	88.01	1	88.01	38.00	0.0001
Residual	32.42	14	2.32		
Lack of Fit	29.6	10	2.96	49.48	0.09
Pure Error	0.24	4	0.060		

**Table 2 molecules-26-03150-t002:** Multiple regression results of the quadratic model.

Factor	Parameter	Coefficient	*t* Value	Standard Error	PC (%)
Intercept	β0	75.45		−	−
A-Time	β1	0.78	0.245	2.401	1.86
B-pH	β2	2.42	28.980	105.843	63.86
C-Concentration	β3	3.29	5.494	24.006	9.48
D-Dose	β4	2.57	−2.751	0.994	4.86
AB	β5	−1.44	−2.405	0.075	0.68
AC	β6	−1.72	−2.455	1.497	0.24
AD	β7	1.35	1.846	7.484	2.25
BC	β8	1.05	12.754	0.748	23.06
BD	β9	0.41	1.102	3.742	0.19
CD	β10	2.500 × 10^−3^	0.839	74.842	13.01
A^2^	β11	1.58	−2.578	0.118	0.80
B^2^	β12	−8.69	−23.612	0.029	21.06
C^2^	β13	2.94	1.120	11.754	3.87
D^2^	β14	3.68	2.960	293.861	0.05

**Table 3 molecules-26-03150-t003:** Kinetic parameters for adsorption of EE2 on Fe_3_O_4_@Ti_3_C_2_.

Kinetic Model	Equation	Parameter	
Pseudo-first-order kinetic model	$Q_{t}=Q_{e}\left(1-e^{k_{1}t}\right)$	Q_e_ (mg g^−1^)	3.0563
		K_1_ (h^−1^)	1.6486
		R^2^	0.9653
Pseudo-second-order kinetic model	$Q_{t}=\frac{Q_{e}^{2}k_{2}t}{1+Q_{e}k_{2}t}$	Q_e_ (mg g^−1^)	3.3003
		K_2_ (g mg^−1^ h^−1^)	0.7709
		R^2^	0.9938

**Table 4 molecules-26-03150-t004:** Isotherm parameters for adsorption of EE2 on Fe_3_O_4_@Ti_3_C_2_.

Isotherm Model	Equation	Parameters	T(K)			
			288	298	308	318
Langmuir	$\frac{c_{e}}{Q_{e}}=\frac{1}{k_{l}q_{m}}+\frac{c_{e}}{q_{m}}$	Q_e_ (mg g^−1^)	6.38	6.25	6.24	5.76
		K_l_ (L mg^−1^)	0.76	0.63	0.86	0.82
		R^2^	0.9940	0.9968	0.9959	0.9971
Freundlich	$lnQ_{e}=lnk_{f}+\frac{1}{n}lnc_{e}$	K_f_ (g mg^−1^ h^−1^)	2.56	2.25	2.67	2.39
		1/n	0.662	0.689	0.641	0.645
		R^2^	0.9693	0.9772	0.9691	0.9729

**Table 5 molecules-26-03150-t005:** Thermodynamic parameters for EE2 adsorption on Fe_3_O_4_@Ti_3_C_2_.

Temperature (K)	InK	ΔG° (kJ mol^−1^)	ΔH° (kJ mol^−1^)	ΔS° (kJ mol^−1^k^−1^)
288	0.5704	−13.65	3.837	0.01799
298	0.5931	−14.69		
308	0.6846	−17.53		
318	0.7080	−18.72		

**Table 6 molecules-26-03150-t006:** Comparison of the EE2 adsorption capacity of Fe_3_O_4_@Ti_3_C_2_ with that of other 2D materials adsorbents.

Adsorbent	pH	Adsorption Capacity (mg g^−1^)	Reference
Entrapped activated carbon in alginate biopolymer	3	0.53	[35]
Multi-walled carbon nanotubes	6	0.47	[36]
4K anthracite	7	1.28	[37]
Biochar	7	2.24	[38]
Fe_3_O_4_@Ti_3_C_2_	6.4	3.83	Present work

**Table 7 molecules-26-03150-t007:** Coded and actual levels of three variables.

Variable	Unit	Notation	Level		
			−1	0	1
Time	h	A	4	6	8
pH		B	4	7	10
Concentration	mg L^−1^	C	0.6	0.8	1.0
Dose	mg L^−1^	D	50	100	150

## Data Availability

All of the recorded data are available in all tables and figures in the manuscript.

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
