# Peer review of "Optimizing Adsorption of 17α-Ethinylestradiol from Water by Magnetic MXene Using Response Surface Methodology and Adsorption Kinetics, Isotherm, and Thermodynamics Studies"

_molecules, 2021, doi:10.3390/molecules26113150_

Round 1

Reviewer 1 Report

Opinion on the paper “Optimizing Adsorption of 17α-Ethinylestradiol from Water by Magnetic MXene Using Response Surface Methodology and Adsorption Kinetics, Isotherm, and Thermodynamics Studies”.

The paper is valuable and worth publishing. In the paper, the synthesis, characterization and application of the composite for the absorption of 17α-ethinylestradiol from aqueous solutions have been described. 17α-Ethinylestradiol is an aquatic pollutant resistant to degradation and it has a negative effect on aquatic animals. Therefore, adsorbents for its removal are sought. The Authors presented magnetic MXene composite Fe3O4@Ti3C2 having a high ability to absorb 17α-ethinylestradiol from dilute aqueous solutions. I agree with the Authors that the interaction between the solid phase of the composite and the 17α-ethinylestradiol probably consists in the formation of hydrogen bonds between the OH groups of the 17α-ethinylestradiol and Fe3O4, while the magnetic properties of the composite serve to easily remove it from the environment through the use of an external magnetic field.

Author Response

We would like to thank the reviewer for giving us  affirmative judgement. Here, we would like to submit our revised manuscript entitled “Optimizing Adsorption of 17α-Ethinylestradiol from Water by Magnetic MXene Using Response Surface Methodology and Adsorption Kinetics, Isotherm, and Thermodynamics Studies (Manuscript ID: molecules-1212629)”. The amendments have been highlighted on the revised manuscript.Thank you for your re-evaluation of our revised submission. 

Reviewer 2 Report

Review Report

Journal: Molecules; Manuscript ID;molecules-1212629; Title: Optimizing Adsorption of 17α-Ethinylestradiol from Water by  Magnetic MXene Using Response Surface Methodology and  Adsorption Kinetics, Isotherm, and Thermodynamics Studies.

This manuscript described the preparation, and characterization, and application (adsorptive removal of 17α-Ethinylestradiol from Water) of  Magnetic MXene (Fe3O4@Ti3C2). Many parts of the paper are well written and provided scientifically sound methods. After a robust evaluation of this manuscript, I think this manuscript can be published in Molecules after a significant revision.

Comments:

  1. The introduction should be elaborated by providing the structure and properties of MXene.
  2. The stability and BET surface area of magnetic MXene (Fe3O4@Ti3C2) would be provided.
  3. The thermal stability and mechanical properties of the magnetic MXene (Fe3O4@Ti3C2) should be demonstrated.
  4. Provide a comparative table of various 2D materials adsorbent, including magnetic MXene (Fe3O4@Ti3C2), to remove 17α-Ethinylestradiol.
  5. A mechanism of adsorption between magnetic MXene (Fe3O4@Ti3C2) and 17α-Ethinylestradiol would be illustrated.

Author Response

We would like to thank the reviewers for giving us constructive suggestions and comments which will help us to improve the quality of our submission. Here, we would like to submit our revised manuscript entitled “Optimizing Adsorption of 17α-Ethinylestradiol from Water by Magnetic MXene Using Response Surface Methodology and Adsorption Kinetics, Isotherm, and Thermodynamics Studies (Manuscript ID: molecules-1212629)”. Our responses are attached below and the amendments have been highlighted on the revised manuscript.Thank you for your re-evaluation of our revised submission. We hope that the revision is acceptable for publication.

Round 2

Reviewer 2 Report

The revised version of this manuscript can be accepted for publication